# Combining Analogy with Language Models for Knowledge Extraction

**Danilo Neves Ribeiro**                    DNRIBEIRO@U.NORTHWESTERN.EDU
**Kenneth D. Forbus**                    FORBUS@NORTHWESTERN.EDU
*Northwestern University, 2233 Tech Drive, Third Floor,*
*Evanston, IL 60208*

## Abstract

Learning structured knowledge from natural language text has been a long-standing challenge. Previous work has focused on specific domains, mostly extracting knowledge about named entities (e.g. countries, companies, or persons) instead of general-purpose world knowledge (e.g. information about science or everyday objects). In this paper we combine the Companion Cognitive Architecture with the BERT Language Model to extract structured knowledge from text, with the goal of automatically inferring missing commonsense facts from an existing knowledge base. Using the principles of distant supervision, the system learns functions called query cases that map statements expressed in natural language into knowledge base relations. Afterwards, the system uses such query cases to extract structured knowledge using analogical reasoning. We run experiments on 2,679 Simple English Wikipedia articles, where the system is able to learn high precision facts about a variety of subjects from a few training examples, outperforming strong baselines.

## 1. Introduction

Knowledge bases (KBs) have proven to be a very useful resource in many artificial intelligence applications including Dialogue, Question Answering, and Semantic Search [Bradeško et al., 2010, Lukovnikov et al., 2017, Ribeiro et al., 2019]. While KBs serve as a good source of explicit knowledge, they often suffer from missing relations and facts. For this reason, extracting knowledge from text with the intent of improving existing KBs has been a long-standing goal in the field of Natural Language Processing. This task is usually referred to as knowledge extraction and is considered challenging because of the inherent diversity and ambiguity of natural language text.

In recent years, there has been substantial progress in automated knowledge extraction, but the resources built by these methods are typically limited by at least one of these four constraints: (1) They lack general world knowledge about common nouns, targeting named entities instead. Examples include NELL [Carlson et al., 2010] and FreeBase [Bollacker et al., 2008] (2) They work with a small set of semantic relations. For instance, WordNet [Miller, 1998] has less than a dozen. (3) They are supervised in nature, requiring vast amounts of training data. Examples include KBP37 [Zhang and Wang, 2015] and TACRED [Zhang et al., 2017]. (4) They represent entities and relations as text strings, which are often ambiguous and harder to manipulate. Examples include ConceptNet [Speer et al., 2017], Aristo Tuple KB [Mishra et al., 2017], ATOMIC [Sap et al., 2019], Dice [Chalier et al., 2020], and Ascent [Nguyen et al., 2021].

By contrast, our goal is to populate an existing ontology, seeded by the facts and schemas already defined in the KB. We use the Companion Cognitive Architecture [Forbus et al., 2017] which has a general-purpose semantic parser capable of generating semantic representations grounded by relations and entities present in the architecture's KB.

In general, KBs containing common-sense world knowledge are incomplete. Consequently, a system has to rely on a small amount of training samples to learn new facts (in our KB around 94.8% of relations are associated with less than 100 facts). To solve this problem, we propose a novel technique based on analogical Q/A training [Crouse et al., 2018]. Our approach efficiently creates *query cases* which are later analogically retrieved and combined to interpret new texts. The system uses distant supervision to automatically generate training data for analogical training which extracts general knowledge from Simple English Wikipedia[1] articles. In this work we combine structural similarity with BERT-guided word sense disambiguation and fact classification for few-shot learning of common-sense facts. To the best of our knowledge, we are the first to combine these two techniques.

## 2. Related Work

Our work relates to learning structured knowledge from natural language text with the goal of extending a KB without relying on labeled corpora. Such tasks are often classified as information extraction (IE) or relation extraction (RE).

Learning under distant supervision for RE without labeled data was initially introduced by Mintz et al. [2009]. They showed how text can be heuristically aligned with a knowledge graph (KG) to generate instances used to train a classifier. Subsequent systems incorporated not only syntactic textual features but also jointly learned KG information to perform extraction, either by syntactic-semantic rules from random walks [Lao et al., 2012], scoring functions of low-dimensional embeddings [Weston et al., 2013], mutual attention between KG and text [Han et al., 2018a], or neural encoder-decoder models [Distiawan et al., 2019]. Other recent methods have focused on using neural networks in the context on few-shot relation extraction [Han et al., 2018b, Gao et al., 2019, Qu et al., 2020, Wang et al., 2021]. Such methods have achieved impressive results on RE but focused on resources that either require many hundreds of training examples or mostly extract facts about named entities. The work of Sharma et al. [2019] is more related to ours as they used distant supervision to learn Cyc style facts [Lenat and Guha, 1993] and were able to leverage the structure of the KB's ontology to improve accuracy. Their experiments however were limited to a small set of relations and they used more shallow textual patterns instead of semantic-parsed interpretation of sentences.

To avoid dependency on pre-defined schemas, Open IE [Yates et al., 2007, Fader et al., 2011, Angeli et al., 2015, Romero et al., 2019, Romero and Razniewski, 2020] extracts knowledge from text using an open set of relations and entities. Commonly, the relation names are represented as text linking two arguments. This representation leads systems to commonly generate multiple instances of the same relation. Riedel et al. [2013] tackled the problem of identifying equivalence between extracted relations by learning canonical schemas. Mishra et al. [2017] used this approach to create a large KB with subject-predicate-

---

1. http://simple.wikipedia.org/

object triples of generic statements about the world. The advantage of our approach over Open IE is that we output symbolic, unambiguous representation of knowledge. Textual representations make reasoning harder, and even though some of the predicates are typed, they often suffer from word sense ambiguity.

Analogy uses the relational structure from semantic representations of texts as a way of learning abstract concepts without vast amounts of labeled data. This approach has been previously applied to solve many natural language problems such as word sense disambiguation [Barbella and Forbus, 2013], question answering [Crouse et al., 2018], multimodal dialog systems [Wilson et al., 2019] and process understanding [Ribeiro et al., 2019]. In our work, analogical similarity is used to retrieve rule-like constructs called query cases. Retrieval of stored cases has been extensively used by many case-based reasoning systems [Daniels and Rissland, 1995, Burke et al., 1997]. However, these case-based reasoning approaches differ from analogical learning since they use shallow pattern matching or standard information retrieval methods instead of structural similarity applied to parsed semantics.

## 3. Problem Definition

The task is to populate an existing KB with facts extracted from natural language text, constrained by the entities and relations already defined in such KB. The standard relation extraction task represents structured knowledge as graphs [Weston et al., 2013, Zhang et al., 2020]. Relations are edges in the graph and are stored as triples $(h, r, t)$, as in (`Spain`,`borders`,`France`). Here, the knowledge is represented using the CycL higher-order logic and OpenCyc ontology [Lenat and Guha, 1993], which is more expressive. Further details on CycL can be found in Appendix A.

Formally, we denote the knowledge base as $\mathcal{KB} = (\mathcal{C}, \mathcal{R}, \mathcal{F})$, where $\mathcal{C}$ is the set of concepts (in CycL they can be either collections, entities, or logical functions), $\mathcal{R}$ is the set of relations, and $\mathcal{F}$ is the set of facts. Each fact $f \in \mathcal{F}$ is a nested tuple, represented as a tree data structure, where the root node $f_1 \in \mathcal{R}$ is the relation and each internal node $f_2, \cdots, f_n$ are the nested arguments. A text corpus is used as input, where each sentence is a sequence of tokens $s = t_1, \cdots, t_m$. The final task is to find all arguments for relations $r \in \mathcal{R}$; that express common-sense beliefs extracted from sentences in the corpus. Note that our proposed task is harder than common RE tasks. They often only attempt to infer either $r$ given $(h, ?, t)$ or $t$ given $(h, r, ?)$, instead of predicting all arguments given a relation.

## 4. Background

We use the Companion Cognitive Architecture as a foundation for our system, as it contains key components that allow for efficient knowledge extraction. Namely, we use the NextKB knowledge base, the CNLU natural language processing system [Tomai and Forbus, 2009], and analogical capabilities provided by SME, the Structure Mapping Engine [Forbus et al., 2017]. We briefly explain each of these components below.

**Knowledge Base:** The NextKB [2] knowledge base integrates material from OpenCyc and FrameNet [Baker et al., 1998], as well as a large-scale lexicon [McFate and Forbus, 2011].

---

2. http://www.qrg.northwestern.edu/nextkb/index.html

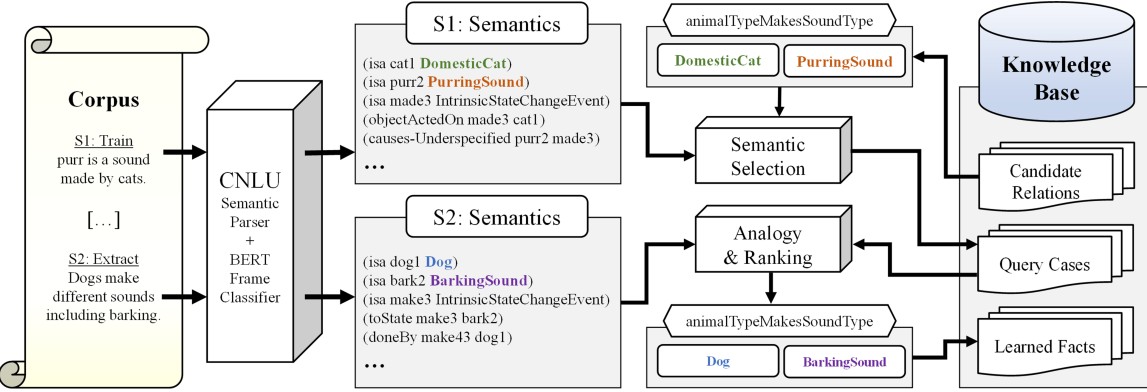

Figure 1: System Overview

All facts in NextKB are represented using the CycL ontology language and they are used as seeds for further inference of missing facts.

**Natural Language Understanding:** The CNLU language system combines Allen's (1995) bottom-up chart parser with ideas from Discourse Representation Theory [Kamp and Reyle, 2013] to produce semantic representations from the input sentences. Specifically, CNLU produces predicate calculus interpretations represented in CycL language. CNLU is task agnostic and the syntactic parse and word sense ambiguities are explicitly represented as distinct and internally consistent choice sets. Choice sets are often derived from FrameNet semantic frames. Logical constraints among the statements are designed such that by choosing one semantic interpretation, other conflicting choices are ruled out.

**Analogy:** We use the Structure-Mapping Engine (SME) by Forbus et al. [2017] to match stored query cases. This component compares two structured, relational representations and creates one or more mappings between them. These mappings consist of (1) a set of correspondences, specifying what entities and predicates in one description go with entities and predicates in the other, (2) a score indicating the similarity between the representations, and (3) candidate inferences that project information from one description to the other. For more computationally efficient retrieval, our approach uses MAC/FAC [Forbus et al., 1995] to apply query cases learned during training for extraction of structured knowledge.

**BERT Language Model:** The Bidirectional Encoder Representations from Transformers (BERT) [Devlin et al., 2018] is a language model trained on a large unlabeled corpus (around 3.3B words from English Wikipedia and BooksCorpus) that jointly conditions on both left and right context for all layers. We use BERT for word sense disambiguation and classification of extracted facts, as described in the following sections. The first input token is always a special token [CLS] which is used for sentence classification while sentence boundaries are marked with the special token [SEP]. The final hidden vector of the special [CLS] token is defined as $T^{CLS} \in \mathbb{R}^H$, and the final hidden vector for the $i^{th}$ input token $t_i$ as $T_i \in \mathbb{R}^H$, where $H$ is the hidden state size.

## 5. Knowledge Base Completion System

Here we describe the knowledge base completion system named *Analogical Knowledge Extraction* (or AKE for short). The system's modules are depicted in Figure 1. The pipeline consists of three steps as described below.

### 5.1 Retrieval of Candidate Relations

During the training phase of the pipeline, the system learns which patterns in a sentence could entail the discovery of new facts. The system uses CNLU to process each sentence from the input corpus, producing predicate calculus semantic representations. Afterwards, the KB is used to retrieve facts that mention more than one concept (collections or entities) which are part of the semantic representation. For instance, the sentence *"Cats use many different sounds for communication, including meowing and purring."* could potentially be paired with (`animalTypeMakesSoundType Cat PurringSound`) while the sentence *"Herbivores only eat plants."* could be paired with (`relationAllExists eatsWillingly Herbivore Plant`). These facts are referred to as candidate relations, represented as the set $\Psi$ where $\Psi_i \in \mathcal{F}$. Their collections and entities are called *anchor concepts* $\Omega$ where $\Omega_i \in \mathcal{C}$.

### 5.2 Word Sense Disambiguation

The CNLU semantic representation may contain conflicting word senses (e.g. *"Mouse"* can be either `Mouse-Rodent` or `ComputerMouse`) and these senses are often derived from FrameNet semantic frames. To disambiguate the word sense choices we use a BERT-based frame classifier that assigns a score to each semantic form given its context.

We fine-tune the BERT language model using a FrameNet annotated data set[3]. Given a sentence with tokens $t_1, \ldots, t_N$, we want to compute the probability of a FrameNet frame $F_k$ being associated with a subsequence of such tokens $t_i, \ldots, t_j$. The model assigns probabilities to each frame resembling the method described in Tan and Na [2019]. During fine-tuning, a new network layer with weights $W \in \mathbb{R}^{K \times 4H}$ is introduced, where $K$ is the number of different FrameNet frames. Finally, the scoring function for a frame $F_k$ is given by:

$$\text{Fra-Pro}\,(F_k) = softmax([T_i : T_j]W^T)_k$$

We assume that the subsequence of tokens is of size four (i.e. $j - i = 4$). Otherwise, the concatenation hidden states $[T_i : T_j]$ are padded with zeroes until it reaches the size $4H$. All choice sets that are not associated with FrameNet frames were assigned a probability drawn from a discrete uniform distribution. This disambiguation module helps the system choose the correct interpretations for both the antecedents and the learned facts.

### 5.3 Construction of Query Cases

Given the set of sentence fact pairs, the system performs a modified version of Analogical Q/A Training [Crouse et al., 2018] to create what we call query cases. These query cases can be thought of as discrete functions that map from a subset of sentence semantic forms

---

3. https://framenet.icsi.berkeley.edu/

(antecedents) to new structured knowledge facts (consequent). Query cases can also be interpreted as discrete functions mapping from input features to output predictions. The main challenge is selecting the relevant subset of the semantics that entails the chosen fact. We use the procedure for semantic selection (Algorithm 1) to automatically choose such subset. This algorithm takes into consideration properties of the semantic representation and anchor concepts to rank the semantic forms that should be added to the query case, namely: semantic connectivity, fact correlation and type generality.

First, the semantic connectivity is used to judge which syntactic and semantic features connect the anchor concepts together, which could potentially identify the relation between such concepts. Second, when there are multiple possible interpretations for the anchor concepts, fact correlation helps with disambiguation by prioritizing concepts that are more closely connected to the candidate relation. Third, type generality gives lower priority to concepts which are too specific since the system needs to generalize the candidate relation to new concepts. The type generality takes as input a concept $X$, and outputs the number of facts in the KB that contain such concept. The formula is as follows:

| **Consequent:** |
| (properPhysicalPartTypes (MaleFn (GeneralizationFn Deer)) (GeneralizationFn Antler)) |
| **Antecedents:** |
| (isa antler6067 Antler) (isa deer6011 Deer) |
| **Abducible Antecedents:** |
| (hasBiologicalSex deer6011 Male) (ownerOfProprietaryThing antler6067 deer6011) (situationIsSuchThat have6030 (resourceAvailable deer6011 antler6067)) (possesses deer6011 have6030) (fe_possession have6030 antler6067) (isa have6030 StaticSituation) |

Figure 2: An example of query case generated from the sentence "Male deer have antlers". Anchor concepts are marked by the logical function GeneralizationFn. Antecedents can be interpreted as input features while consequent as output predictions

$$\text{Gen}(X) = |\{F_x \mid F_x \in \mathcal{F} \ \wedge \ X \in \ F_x\}|$$

The subroutine Ont-Con was introduced by Ribeiro et al. [2019] and computes the ontological similarity of two expressions. Given two collections, entities, or relations $\varphi_1, \ \varphi_2 \in \{\mathcal{C} \cup \mathcal{R}\}$ in the KB, and the set of structural relations of the ontology (examples in Open-Cyc are `isa`, `argIsa`, `genls`), form a graph between $\varphi_1$ and $\varphi_2$. Let $P$ be the set of paths between $\varphi_1$ and $\varphi_2$, $e_{u,w}$ be the vertices in each of these paths and $deg(x)$ be the out-degree of a vertex $x$. Then traversing the nodes in this graph can give the relatedness between two concepts. Therefore, we define the ontological connection to be:

$$\text{Ont-Con}(\varphi_1, \varphi_2) = \sum_{p_w \ \in \ P} \frac{\prod_{e_{u,w} \ \in \ p_w} deg(e_{u,w})^{-1}}{|P|}$$

The semantics selection algorithm filters out many of the false positive candidate relations that have low scores. Once the subset of semantic forms is selected, the query case is created. One example of such query case generated by the sentence *"Male deer have antlers."* is depicted in Figure 2. All query cases created during the training phase are stored in the KB for subsequent retrieval.

## 5.4 Learning New Facts

When faced with natural language text, the system must extract structured knowledge that could be expressed by the known relations from the training data. The first step is to parse

---

**Algorithm 1:** SEMANTICS-SELECTION

---

**Input:** Sentence semantics $\Upsilon$, retrieved candidate relation $\Psi$, anchor concepts $\Omega$
**Output:** Subset of semantics $\Upsilon^* \subseteq \Upsilon$
**Parameters:** $\lambda_1, \lambda_2, \lambda_3, \lambda_4, \varepsilon_1$
First select connecting forms $\upsilon_{u,v} \subseteq \Upsilon$ from paths between any two anchors $\Omega_u$ and
  $\Omega_v$ through the semantic representation graph with depth up to 3;
**foreach** $\Upsilon_u \in \Upsilon$ **do**
   | **if** $\Omega_v \in H_u$ **then**
   |    | $con \leftarrow |\upsilon_{v,x}|; \; fac \leftarrow \sum_v \text{ONT-CON}(\Omega_v, \Psi); \; gen \leftarrow \text{GEN}(\Omega_v);$
   |    | $scr[\Upsilon_u] \leftarrow \; \lambda_1 * con + \lambda_2 * log_{10}(1 + fac) + \lambda_3 * log_{10}(1 + gen)$ ;
   | **else**
   |    | $\Omega' \leftarrow$ all collections from $H_u$;
   |    | $scr[\Upsilon_u] \leftarrow \lambda_2 * \sum_v \text{ONT-CON}(\Omega'_v, \Psi);$
   | **end**
   | $scr[\Upsilon_u] \leftarrow scr[\Upsilon_u] + \lambda_4 * \text{FRA-PRO}(\Upsilon_u);$
**end**
$\Upsilon' \leftarrow$ sort $\Upsilon$ by $scr[\Upsilon_u]; \; \Upsilon^* \leftarrow \emptyset;$
**foreach** $\Upsilon'_u \in \Upsilon'$ **do**
   | **if** *not* $\text{CONFLICTS}(\Upsilon'_u, \Upsilon^*_v)$ *and* $scr[\Upsilon_u] > \varepsilon_1$ **then**
   |    | $\Upsilon^* \leftarrow \Upsilon^* \cup \Upsilon'_u$ ;
   | **end**
**end**

---

the text using CNLU, which generates the semantic representation of the sentences. Such semantic representations are used as probes for analogical retrieval using MAC/FAC and further processed using SME, which returns a small number of query cases stored during training that best matches the semantic representation of the sentence (up to 20 cases per probe).

**Ontological Scoring:** In order to decide if the retrieved cases contain sufficient evidence that the semantics of the sentence entails a new fact, the system uses a heuristic to compute scores for each instantiated case. This ontological scoring ensures that the learned facts are relatively similar to the original training facts, that the word senses of the semantic interpretation have high probability and that the concepts are not too generic.

Given the sentence semantics $\Upsilon$ and the instantiated query case with antecedents $A$ and consequent $C$, anchors of the instantiated consequent $\Omega^c$ and the anchors of the original query case consequent $\Omega^o$. Then the score is given by:

$$\lambda_5 * \sum_{A_u \in A} \text{FRA-PRO}(A_u) + \lambda_6 * |A| - \lambda_7 * log_{10}(\text{GEN}(\Omega^c) +$$

$$\lambda_8 * log_{10}\left(\sum_{u,v} \text{ONT-CON}(\Omega^c{}_u, \Omega^o{}_v)\right)$$

The system uses this score to filter out new facts that are below a hyper parameter threshold $\varepsilon_2$. The new facts, the provenance and the scores are all stored in the KB.

**BERT Fact Classification:**   The second classification function takes advantage of BERT's contextual representation and word embeddings to capture common sense knowledge which were implicitly stated in its large pre-training corpus. Inspired by the approach used by Yao et al. [2019], we fine-tuned BERT to predict the plausibility of extracted facts. For BERT input, we represent concepts and relations by an automatically constructed textual format. For instance, the relation (`pathogenCausesConditionType Borrelia LymeDisease`) is converted into *"[CLS] pathogen causes condition type [SEP] lyme disease [SEP] borrelia [SEP]"* which is used as input to the model.

To create negative training examples, we simply switch a concept in a positive training example from $\mathcal{F}$ with some other randomly chosen concept from $\mathcal{C}$. We also create negative examples by shuffling the order of concepts inside the fact (excluding facts in the set of symmetrical relations $\mathcal{R}^\circ$). Considering a simplified representation of a fact as a triple $(h, r, t)$ where $r \in \mathcal{R}$ and $h, t \in \mathcal{C}$, the set of negative training examples is defined by:

$$\mathcal{F}^- = \{(t, r, h) | (h, r, t) \in \mathcal{F} \wedge r \notin \mathcal{R}^\circ\} \cup \{(h, r, t') | t' \in \mathcal{C} \wedge t' \neq t \wedge (h, r, t') \notin \mathcal{F}\}$$
$$\cup \{(h', r, t) | h' \in \mathcal{C} \wedge h' \neq h \wedge (h', r, t) \notin \mathcal{F}\}$$

When fine-tuning BERT we use both $\mathcal{F}$ and $\mathcal{F}^-$ as training data. A new network layer with weights $W' \in \mathbb{R}^{2 \times H}$ is introduced. The scoring function $s^\tau$ for a fact $\tau \in \mathcal{F} \cup \mathcal{F}^-$ is given by the formula: $s^\tau = sigmoid(T^{CLS}W'^T)$ where $s^\tau \in \mathbb{R}^2$ is a two-dimensional vector representing the model output. The pre-trained parameter weights and new weights are updated using gradient descent that minimizes the cross-entropy loss:

$$L = \sum_{\tau \in \mathcal{F} \cup \mathcal{F}^-} (y^\tau log(s_0^\tau) + (1 - y^\tau)log(s_1^\tau))$$

Where $y^\tau$ is the output label defined by $y^\tau = 1$ if $\tau \in \mathcal{F}$ and $y^\tau = 0$ if $\tau \in \mathcal{F}^-$. After training, this model achieves a validation accuracy of 89.0% on the task of fact classification. In following experiments, the BERT fact classification is applied after the ontological score for further filtering of learned facts.

## 6. Experiments

### 6.1 Dataset and Corpus

In order to evaluate our system, we experimented with extracting knowledge from Simple English Wikipedia articles. We select a subset of the articles by compiling a list of common nouns, and filtering out articles with titles that are not part of this list. The final corpus contains 2,679 articles.

We also select a subset of the NextKB knowledge base which expresses general facts about the world. This was done by selecting facts contained within relevant micro-theories (term in Cyc used to describe collections of concepts and facts) such as `AnimalActivitiesMt`, `BiologyMt` or `HumanActivitiesMt`. Relations which we considered not proper for extraction (e.g. `quotedIsa` and `givenNames`) were also removed. The final data set is publicly available[4] and contains 66,649 facts with 3,745 distinct relations, covering 21,462 concepts.

---

4. https://github.com/dnr2/analogical-ke

| Method | Estimated Precision |
|---|---|
| Relation Extraction (CNN) * | 17,1% |
| Relation Extraction (BERT) * | 20,8% |
| Text-to-Text (T5) | 26.0% |
| Analogy (AKE) | 45.7% |
| Analogy (AKE) + BERT fact classifier | **71.4%** |

Table 1: Evaluation results showing the estimated precision of extracted facts from the Simple English Wikipedia corpus. Models with * only work on triples, where relations only allow for two arguments.

Knowledge bases representing generic facts are often incomplete, where the frequency of relations follows a long tail distribution [Xiong et al., 2018]. Our data set also follows this pattern as shown in Figure 3, which further justifies the use of data efficient learning methods such as analogical learning for this type of task.

### 6.2 Baselines

We compare our system against three baselines. The first is based on the Text-to-Text Transfer Transformer (T5) [Raffel et al., 2019], which is fine-tuned to take as input a sentence and output a sequence of structured knowledge facts. For instance, the sentence *"Books are made of paper."* and the structured fact *"typicalMainConstituent-TypeType [S] BookCopy [S] Paper"* were used as input and output pairs during training.

The other two baselines are Relation Extraction models from OpenNRE [Han et al., 2019], namely the CCN [Zeng et al., 2014] and BERT [Devlin et al., 2018] based encoders. We adapt the task by excluding any facts with more than two arguments. The surface form of the concepts in the KB (taken from CNLU lexical mappings) are used to identify possible head and tail arguments. The Relation Extraction models use these head and tail concepts as inputs and predict the relation among them. Appendix B contains further details about training and hyper parameters.

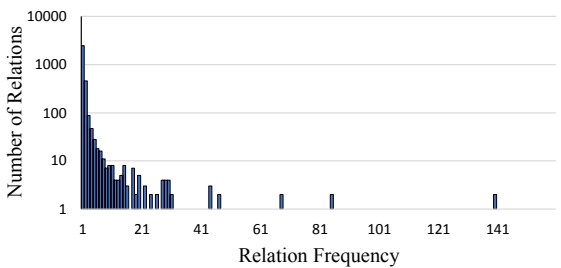

Figure 3: A histogram showing the distribution of relations according to their frequency (number of occurrences in facts) in the data set. Evidence that most relations are associated with a few facts.

### 6.3 Evaluation and Results

After running each system on the sentences from Simple English Wikipedia we evaluate the learned facts. Notice that all extracted facts that were already present in the KB were discarded. Often the evaluation of relation extraction systems is done by creating a held-out test set [Mintz et al., 2009, Han et al., 2018a]. This would not be appropriate for this data set since the KB is very sparse and has many missing facts. For this reason, we opted for manual evaluation of the results. We randomly sampled 8% of the extracted facts which

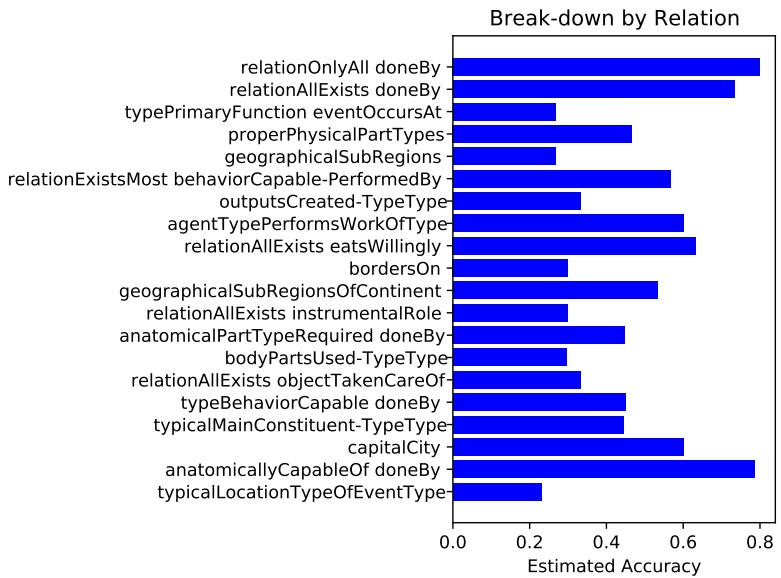

Figure 4: Estimated accuracy of a subset of facts extracted by the *AKE + Bert fact classifier* system, broken-down by different relations.

were given to a graduate level annotator for manual evaluation. We report the estimated precision for each compared method in Table 1.

The Text-to-Text (T5) was the best baseline, with estimated accuracy of 26.0% and able to extract 6772 new facts. The adapted Relation Extraction models with CNN and BERT encoders extracted 2,448 and 2,804 new facts, respectively. Interestingly, these models performed fairly well on the distant supervision development set (achieving over 80% F1 scores), but did not perform as well when it come to discovering facts not already in the knowledge base, possibly because such facts are outside of the distribution of the training data. Potential improvements to these baseline models include using better word senses for aligning sentences with facts or learning KG embeddings [Distiawan et al., 2019, Wang et al., 2021].

The purely analogical system has an estimated accuracy of 45.7% and is able to extract 4,458 facts covering 58 distinct relations. The accuracy improves considerably after applying the BERT fact classification, jumping to 71.4%. Figure 4 shows the precision break-down for a subset of the relations (up to 30 extractions per relation). After inspecting the results more closely, we noticed how the system is capable of performing well even in the case of sparse training data. For instance, the relation (`relationAllExists eatsWillingly ?x ?y`) only has four instances in the NextKB set of facts. However, the system is able to expand that to 156 new facts. We investigated the inaccuracies of the Analogy system and found that roughly 42% of the failures were related to semantic parsing errors. The remaining errors were mostly due to errors in the BERT-based disambiguation component (which were often caused by the lack of training data from FrameNet) or the incorrect semantic selection during the creation of query cases.

## 7. Conclusion

With the intention of populating large common-sense KBs, our work introduced a pipeline for extracting structured knowledge from a text corpus. While learning instance level knowledge about named entities (e.g. companies or persons) is important, we believe that learning type-leveled information is more relevant for building general purpose AI systems. We have presented a hybrid approach built on the Companion Cognitive Architecture that combines analogical learning and BERT-based word sense disambiguation and fact classification. We show how this approach is effective even when applied to sparse KBs, where the frequency of relations follows a long-tail distribution. The system was able to outperform strong baselines and learn new facts with reasonable accuracy from a small set of distantly supervised training examples, expanding the number of facts for certain relations by two orders of magnitude.

Some directions for future work include (1) Processing more data and extracting more facts from the web. Simple English Wikipedia has served as a starting point for showing the effectiveness of our approach, but to significantly increase the coverage of an existing KB a larger scale text corpus would be required. (2) Extracting richer representations of knowledge. We hypothesize that our system is suitable for learning and identifying knowledge represented as frames, since it can learn facts with arbitrary arity and depth. For instance, this could be useful for extracting knowledge about events or processes. (3) Exploring more statistical properties of query cases. We believe that identifying which subset of antecedents contributes most for a given relation could improve the system's accuracy.

### Acknowledgments

This research was supported in part by the Machine Learning, Reasoning, and Intelligence Program of the Office of Naval Research (Grant No. N000142012447). We thank the members of the Northwestern Qualitative Reasoning Group for their insightful comments and suggestions.

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

## Appendix A. CycL Representation

The key differences between CycL representation and other commonly used representations are (1) CycL represents concepts as disambiguated constants. For example, the concept *"Mouse"* can be represented by either `Mouse-Rodent` or `ComputerMouse`, depending on the word sense. Concepts are represented by collections, e.g. `ComputerMouse` is a collection, while `Mouse32` might denote an instance of that collection, a specific computer mouse; (2) relations can be of arbitrary arity, instead of only using binary relations, which allows relationships like `between` to be represented; (3) relations can be higher-order, e.g. `(relationAllExists eatsWillingly Omnivore Meat)` takes predicates and concepts (aka `Collections` in OpenCyc) as arguments; (4) logical functions can be used to produce new terms, e.g. `(FruitFn AppleTree)` represents the collection of the fruits of apple trees.

## Appendix B. Training and Hyperparameters

Follows the training details, including choice of Hyperparameters for the system sub-components and neural models which were described previously.

**Word Sense Disambiguation:** The word sense disambiguation trained on the FrameNet data. The BERT base and uncased model was fine-tuned using the Adam optimizer, with a learning rate of $1 \cdot 10^{-4}$, trained for 10 epochs.

**Semantic Selection and Ontological Scoring:** These modules used a linear combination of scores to select the best semantic representation, as well as scoring learned facts. The values for the parameter as as follows: $\lambda_1 = 1, \lambda_2 = 0.2, \lambda_3 = 5, \lambda_4 = 5, \varepsilon_1 = 0.3, \lambda_5 = 3, \lambda_6 = 1, \lambda_7 = 3, \lambda_8 = 3, \varepsilon_2 = 2.5,$. These values were chosen depending on the importance of each score, taking into consideration the results on a sample of the training data.

**Fact Classifier:** The fact classification model fine-tuned a BERT (base uncased) using a learning rate $5 \cdot 10^{-5}$, AdamW optimizer and trained for 3 epochs. The generated dataset was split into 63,805 training and 7,090 validation data-points.

**Text-to-Text:** When training the text-to-text model we fine tune T5-small on 3,962 sentence-fact pairs using learning rate $3 \cdot 10^{-4}$, AdamW optimizer and trained for 2 epochs.

**Relation Extraction:** The relation extraction models are trained on 28,800 training and 3,000 validation sentence-fact pairs, %50 of which were negative examples. Negative examples (labeled as "not_related") were created using sentences that reference two entities that are not connected by a fact in the KB. All facts with more than two examples were discarded. The model with CNN encoder was trained with a learning rate of $1 \cdot 10^{-1}$, and 160 batch size for 500 epochs. The BERT based model was trained with a learning rate of $2 \cdot 10^{-5}$, and 64 batch size for 10 epochs. The remaining parameter were kept as the OpenNRE defaults.