# OpenReview forum: "Combining Analogy with Language Models for Knowledge Extraction"
_AKBC.ws/2021/Conference — AKBC 2021_

### Official Review · Reviewer_ZXfU · 2021-07-20
**Interesting work, but confused about some details**

**Rating:** 6
**Confidence:** 3

**Review:**

This paper presents an approach for augmenting a knowledge base with general knowledge facts by extracting them from text. The system parses new sentences, converts them into semantic relations, ties them to the existing knowledge graph to compute a connectivity with the current knowledge base, and uses BERT to evaluate plausibility. An experiment shows that the precision of the proposed system in extracting facts is quite high, higher than the baselines.

My concerns are mostly on the clarity of the paper, and inquisition about additional experiments.

Quality: There are some experimental details missing, and some experiments which would have been interesting to see. First, how many total facts were extracted in the different systems (it seems about 4.5k for the main system, but other systems are not mentioned)? Because the system is trained (as I understand) by constantly building the knowledge graph (adding new facts as it processes text), evaluating across multiple random orderings would be useful to see how robust it is to this. Also, the required quality of the initial KB is not evaluated -- how does resulting precision change as the initial size of the KB changes?

Is there some way to estimate recall? For example, by annotating a few Wiki pages and seeing if the system can capture all of the facts?

What was the BERT fact classification model trained on?

Clarity: There were many details that I was not familiar with which I would have liked elaboration on. For example, I didn't know what Gen(X) was supposed to be computing, or what F and X were. It seemed like the "sentence semantics" was the same as the CNLU output, but I'm not sure. The indices in the facts in Figure 2 were never explained. When referring to existing work, it would help to elaborate a bit on what the inputs/outputs are, and a brief summary of the method (e.g., Q/A training).

When is word sense disambiguation applied?

I was pretty confused about what exactly a query case is and how it is related to analogies. I'm not sure if there is an example in the paper besides Figure 2, which doesn't have much detail in the caption.

Originality: I'm not familiar with other work that does something similar to this.

Significance: The results are promising, and I appreciate the discussion on error types (although it would be interesting to see examples and proportions of errors in the different categories).

---

### Official Review · Reviewer_vVFY · 2021-07-21
**Interesting method for extracting complex knowledge structures by using pre-trained LMs with existing IE pipelines.**

**Rating:** 7
**Confidence:** 4

**Review:**

Summary:
The paper presents a method for combining the Companion Cognitive Architecture with BERT for the purpose of discovering new Cyc-style common-sense facts. By using BERT at relevant points in the Companion pipeline, the paper achieves a significant performance improvement over an end-to-end T5 model. In particular the BERT network for filtering facts proposed by the cognitive systems results in a huge jump in precision from 46 to 71.

Strengths:
1. This work addresses a challenging problem of extracting new Cyc-style facts from text. Using end-to-end Seq2Seq architectures fail to achieve good performance due to the complexity of the extracted knowledge. Hence, the idea of using prior pipeline based architectures with neural components introduced at appropriate points seem to be a good way of tackling these complex problems.
2. Using the pre-trained knowledge of LMs for filtering incorrect facts proposed by existing systems can take advantage of other ever-increasing pre-trained LMs and applied to other knowledge extraction tasks as well.

Weaknesses:
Since the paper aims to extract new facts to complete a commonsense based KB, they should have considered contemporary commonsense KBs like Atomic (Sap et al., 2019) or ConceptNet (Speer et al., 2017). Another alternative is to demonstrate that OpenCyc KB gives some benefits that are absent in the above KBs.

Questions:
1. It is not clear to me what parts of the Analogy system are borrowed from Companion Cognitive Architecture and how much of it is your novelty? Could you please elaborate on this?
2. In the T5-baseline, how many generated predictions are incorrect because of generating entities or relations that are outside the KB scope? Without constraining the output in some manner, it does seem difficult for the T5 model to achieve reasonable performance.
3. Why has this technique been applied only to Simple Wikipedia? What problems do you foresee on applying this to more complicated, general text? Does the semantic parsing involved degrade in performance?
4. It is not clear how exactly you are applying Relation Extraction models for the task. Since both the arguments and relations need to be extracted, how do CNN or BERT based classifiers achieve this? Do you assume the arguments are already given in these cases?
5. What is the use of Ontological Scoring in the entire pipeline? What does it do that BERT Fact Classification fails to achieve?

---

### Official Review · Reviewer_tP1W · 2021-07-22
**Nice ideas, pipeline approach combining existing IE approaches for extraction with BERT for filtering.**

**Rating:** 6
**Confidence:** 2

**Review:**

This paper designs a method to extract structured information in the form of commonsense facts from unstructured text. They combine existing building blocks with BERT based modules to provide a novel method that extracts facts with high precision even when applied to KBs that are quite sparse.
Overall, the paper makes a valuable contribution. However, it is a bit hard to follow without having extensive background in this area and can be re-organized to be easier to follow.

Strengths:

1) Expands on prior work in RE that usually only does tail/relation prediction to extracting the entire relation tuple.
2) Useful tricks to overcome problems such as word sense ambiguity by using BERT as an additional module for frame classification and filtering extracted facts by leveraging the power of contextualized representations pre-trained on large amounts of data.

Weaknesses:

1) The authors expect quite a lot of background knowledge of the area when describing the method and the paper would benefit from a little more explanation of key terms and components used.
2) In addition to the precision, it would be good to have a sense of recall on the selected subset of articles without which it is a bit hard to evaluate.

Minor comments:

The algorithm 1 is hard to follow, maybe it would be easier if the variables used had some surface level similarity to what they refer to? Given so many symbols, it’s hard to keep track while reading the algorithm. A lot of the terms in the algorithm are only described in the page following the one which actually refers to the algorithm, making it impossible to make sense of unless you skip ahead. Re-organizing these sections so that the Algorithm 1 comes after describing the parts might also help with clarity.

Questions:

1) For fact classification, how is the validation set constructed? How big is it and how many types of relations does it cover?
2) Ontological scoring filters facts that are not similar enough to existing facts in the KB. Does this place an implicit upper bound on the types and number of facts that you can extract as a function of the KB that is used to initialize this process? It would be useful to know statistics on the minimum number of facts and concepts, as well as a distribution over their types, that is needed to obtain good performance using this approach.
3) Maybe it is also possible to combine this with knowledge from wordnet that has information about synonyms and hypernyms, to extract further facts that are implied by the base facts that are extracted by this model?
4) Since the goal of this work was to populate large commonsense KB and given that this paper has shown proof of viability of the approach, would it be possible to apply your approach to add more facts to existing commonsense knowledge bases like ATOMIC?


Cosmetic comments:

1) Change all open quotations to be `` as used in Latex.
2) T5 is “Text to Text Transfer Transformer”, not “Sequence to Sequence Transfer Transformer”
3) Cite first occurrence of “Cyc style facts” in related work (or forward reference this or explain what it is.)
4) Section 4 Companion cognitive architecture -> Companion Cognitive Architecture

---

### Decision · Program_Chairs · 2021-08-17

**Decision:**

Accept

**Comment:**

This paper proposes a method for extracting structured commonsense facts from natural language text. They show that end2end seq2seq architectures fail on this task. Their pipeline approach with Companion Cognitive Architecture to generate candidates in combination with BERT fine-tuned for rejecting bad candidates resulted in high precision extractions. In general, their proposal of "using the pre-trained knowledge of LMs for filtering incorrect facts proposed by existing systems" is applicable to any knowledge extraction tasks and upcoming better and better pre-trained LMs. Which makes this paper attractive for the AKBC community. Reviewers have made an excellent set of suggestions that can be incorporated to make the draft clear and more accessible to people new to this area. Also, multiple reviewers suggested adding recall estimates (based on a smalls et of articles) which can help strengthen the paper.